# Numerical Modeling of Damage Caused by Seawater Exposure on Mechanical Strength in Fiber-Reinforced Polymer Composites

**DOI:** 10.3390/polym14193955

**Published:** 2022-09-22

**Authors:** Hugo Vidinha, Ricardo Branco, Maria Augusta Neto, Ana M. Amaro, Paulo Reis

**Affiliations:** CEMMPRE, Department of Mechanical Engineering, University of Coimbra, Rua Luís Reis Santos, Pinhal de Marrocos, 3030-788 Coimbra, Portugal

**Keywords:** seawater exposure, mechanical properties, fiber-reinforced polymer, numerical modelling

## Abstract

Fiber-reinforced polymer composites are frequently used in marine environments which may limit their durability. The development of accurate engineering tools capable of simulating the effect of seawater on material strength can improve design and reduce structural costs. This paper presents a numerical-based approach to predict the stress–strain response of fiber-reinforced polymer composites exposed to different seawater immersion times, ranging from 0 to 900 days. A three-dimensional numerical model has been implemented using a static implicit finite element analysis along with a user-defined material (UMAT) subroutine. Puck’s failure criterion was used for ultimate failure analysis of the laminates, while Fick’s first diffusion law was used to predict the seawater absorption rate. Overall, the simulated stress–strain curves were close to those obtained experimentally. Moreover, the model agreed well with the experimental data regarding the maximum stress and the strain at failure leading to maximum errors lower than 9% and 11%, respectively. Additionally, the simulated strain fields agreed well with the experimental results measured by digital image correlation. Finally, the proposed procedure was also used to identify the most critical surfaces to protect the mechanical components from marine environments.

## 1. Introduction

Fiber-reinforced polymers (FRP) possess very attractive physical and mechanical properties, such as good fatigue resistance, high specific stiffness, high specific strength, and easy construction relative to traditional materials [1]. This is reflected in the growth of their use in areas where light and strong structures are required, for example, in the aerospace and automobile industries. The commonly used FRP composite reinforcements are made from carbon fiber, glass fiber, basalt fiber and aramid fiber [2]. Glass fiber-reinforced polymers (GFRP) are the most frequently used FRPs mainly due to their low cost and relatively good tensile strength [3]. On the other hand, basalt fiber reinforced polymers (BFRP) are more expensive but offer improved specifications such as higher strength and chemical resistance [4]. In the case of aramid fiber reinforced polymers (AFRP), they are not frequently used in structural applications due to low compressive strength and high cost. However, they are one of the best choices for ballistic protection applications [5]. Lastly, carbon fibers possess exceptional mechanical properties having high specific strength, low density, and also good electrical properties, namely excellent conductivity [6]. As a result, although carbon fiber-reinforced polymers (CFRP) are high-priced, they are also widely used in applications requiring high specific stiffness, enhanced fatigue properties and high specific strength, including aerospace, military and wind power applications [7,8]. In addition, due to their good corrosion resistance, FRPs have aroused great interest in multiple marine applications, namely in ship hulls, sonar domes, propellers, railings, submarine structures and offshore sectors [9,10,11,12]. However, although FRPs have been used for over 50 years in marine structures, their adoption frequently requires high design safety factors, since long-term durability predictions are generally based on experience [13]. This can be partly justified by the fact that the usage of composite materials in hostile environments can significantly reduce their mechanical behaviour.

Extensive research has been conducted to evaluate the effects of hostile environments on composite materials [14,15,16,17,18,19,20]. The absorption of water is detrimental to the mechanical properties of FRP composites, since it affects the resin matrix by plasticization and also deteriorates the matrix–fiber interface [21,22,23,24]. However, the degradation of the matrix resin due to water absorption is, in general, less relevant than the degradation of the matrix–fiber interface. In fact, several studies have demonstrated that the bonding between the matrix and fiber is the predominant factor causing the degradation of the composite [21,25,26,27,28]. Furthermore, Apicella et al. [29] demonstrated that water can also hydrolyze the resin [30]. The same study showed that isophthalic resins revealed the lowest hydrolytic stability, while, on the contrary, vinyl ester resins revealed the highest stability after exposure. 

Temperature changes also play an important role in mechanical behaviour of composite materials. Netravali et al. [31] studied the effect of water sorption at different temperatures on the reversibility of the plasticization and concluded that, at ambient temperature, the plasticization was mostly reversible. However, the samples exposed at 70 °C revealed a considerably lower glass transition temperature. In another study, Wong et al. [32] addressed the effect of multiple sorption cycles in a glassy epoxy resin. It was found that anomalous sorption cycles are verified not as a result of the development of cracks but also due to an alteration in resin chemical structure caused by an oxidation reaction. Ellis et al. [33] investigated the influence of water in epoxy resins and concluded that even a small amount of water uptake (e.g., 1–2 wt%) has a considerable plasticizing action, leading to degraded properties.

As described above, the damage induced by seawater can emerge in numerous forms. Analogously, water absorption in composite materials is also a complex phenomenon, which many causes can influence, such as the resin and curing agent, fabrication, fiber volume fraction, orientation of reinforcement, area of exposed surfaces, and temperature, among others [27,34,35]. Water penetrates the structure mainly via diffusion, through the resin matrix [36,37], and most water absorption is absorbed in the matrix. However, Tsenoglou et al. [38] evaluated the water penetration of an imperfectly bounded fiber–matrix interface and found that, essentially in low quality composites, the micro-channels surrounding the fibers can be the predominant conduits of moisture diffusion. Furthermore, voids can be developed during the composite cure process, which also increases the water absorption during the FRP life cycle [39]. Weitsman et al. [40] analysed the microcracking influence on the water absorption behaviour in polymers and found that micro-cracks increase the absorption capacity and accelerate the diffusion process. Although microcracking is extensively linked with the increase in absorption capacity, it is still highly random and difficult to evaluate [40]. Pritchard and Speake [41] studied the kinetics of moisture absorption on GFRP for temperatures between 30 and 100 °C and found that the saturation point is reached more quickly at higher temperatures. However, the maximum water absorption remained unaltered. The same study showed that the residual mechanical properties were found to be functions of true absorbed water content.

Saadatmanesh et al. [42] studied the effect of different aggressive environments on the tensile properties of FRPs. An interesting outcome was that the carbon FRPs, when exposed to most environments, revealed excellent durability. Even after 27 months of exposure, the mechanical properties of the specimens were minimally affected by these environments. However, the exposure effects were significantly higher in the glass FRP specimens. After 20,000 h of exposure, i.e., about 830 days, the ultimate normalized strength of the carbon-fiber specimens decreased by around 10%, while the glass–fiber ones decreased by 30 to 40%. In another study, Bian et al. [35] verified that the tensile strength of glass FRPs was reduced by 13% after only 42 days of immersion. Similarly, Gellert and Turley [43] conclude that the flexural strength continued to degrade for GFRPs as water uptake continued toward saturation. Moreover, the authors reported strength losses between 15 and 21% as well as a decrease in the interlaminar shear strength by 21% after 485 days of immersion.

Despite the detrimental effect of hostile environments on mechanical behaviour of FRPs being well documented in the scientific literature, the development of numerical tools capable of correlating the exposure time with the mechanical performance is much less common. Gholami et al. [44] conducted a micro-scale finite element analysis concerning the hydrothermal degradation of FRP. However, the scope of the study was limited to predicting the properties of the composite using representative volume elements. More recently, Feng et al. [45,46] addressed the effect of hydrothermal environments, via both experimental and numerical approaches, using a multiscale modelling method. Nevertheless, the effects of permeable and non-permeable surfaces on mechanical behaviour of immersed structures were not considered.

Therefore, although the development of the above-mentioned numerical tools are a major challenge, from a perspective of engineering design, such tools can help to improve the overall design of marine structures, reduce safety factors, partly reduce the expensive and time-consuming experimental testing campaigns, and speed up the time to market. With the advent of computer technology, advanced numerical methods have been a strong ally in the analysis of complex phenomena. Due to its flexibility, and because it is widespread in current engineering industry, the Finite Element Method (FEM) is the ideal choice to devise this tool.

In this context, this paper aims to develop a three-dimensional (3D) numerical model capable of predicting the effect of seawater on mechanical strength of glass fiber-reinforced polymers (GFRP) using the FEM. Puck’s failure criterion, a failure theory for unidirectional fiber-reinforced polymer composites, was applied to define the ultimate failure of the laminates. Then, Fick’s first diffusion law was used to predict the seawater absorption rate. Finally, a relation between the Fick’s law and the diffusion distance was introduced to predict the seawater concentration into the material. The numerical model was implemented in the ABAQUS commercial software by developing a user-defined material (UMAT) subroutine. After that, the Digital Image Correlation (DIC) technique was used to validate the implementation of Puck’s theory by providing full-field strain measurements. Finally, the predictive capabilities of the proposed approach were compared with experimental tensile test results obtained for different seawater immersion times ranging from 0 to 900 days.

The first part of this paper features the materials and the procedures used to validate the numerical models. The second part outlines Puck’s failure criterion, the proposed numerical model, and their implementation in the subroutine. After that, the numerical simulations are compared to the experimental stress–strain test results performed for immersed and not immersed glass fiber-reinforced polymer composite laminates. Finally, the proposed procedure is used to improve structural design by applying impermeable coatings.

## 2. Material and Experimental Methods

Composite glass/epoxy laminates were used in this research. The laminates were produced using 1195P (195 g/m^2^) glass fiber and Biresin^®^ (Sika Services AG, Zurich, Switzerland) CR122 epoxy resin with the Biresin^®^ CH122-3 hardener. The amount of hardener was 30% of the resin’s total weight, as suggested by the manufacturer. The laminates were placed in a vacuum bag and compressed with a load of 2.5 kN for 12 h to preserve the original thickness. During the first 4 h, the vacuum bag remained connected to a vacuum pump to eliminate air bubbles. The post-cure was conducted in agreement with the manufacturer in an oven at 60 °C for 8 h. The laminates were composed by 12 plies distributed according to the layout [0°, 45°, 90°, 45°, 0°, 90°] s and had an overall dimension of 330 mm × 330 mm × 2.3 mm.

The specimens were cut from the laminate panels with dimensions of 165 mm × 22.5 mm × 2.3 mm. Then, a central hole measuring 5 mm in diameter was drilled using a twist drill bit operated at a suitable cutting speed to prevent delamination. After the manufacturing process, three specimens were immersed in seawater for 900 days, and another three were used as control samples in a dry condition, i.e., without immersion in seawater. The former group was weighted regularly during the immersion stage to evaluate the water absorption rate. In total, six tensile tests were conducted, three were used as control samples (i.e., not immersed in sea water), and the other three were used to evaluate the effects of seawater (i.e., immersed 900 days in seawater). The tensile tests were performed at room temperature, according to the ASTM D3039 standard, with a displacement rate of 2 mm/min. A Shimadzu Autograph AGS-X (Shimadzu Scientific Instruments, Kyoto, Japan) machine with a load capacity of 150 kN was used. The stress–strain response was acquired from a 25 mm-long mechanical extensometer connected directly to the specimen using the Trapezium X software (version 2.2).

The strain fields at the specimen surface were obtained using Digital Image Correlation (DIC) using a VIC-3D system (Correlated Solutions, Irmo, SC, USA). The specimens were previously prepared and painted with a white matte base coat, and a random speckle pattern of black paint was applied to create a pattern on the center zone of the specimen. To capture the strain field, two high-velocity cameras (Correlated Solutions, USA) were placed at approximately 30°, symmetrical in relation to the specimens, at the maximum resolution of 1624 pixels × 1224 pixels. During the loading stage, images were captured using these two cameras, and strain fields were obtained with specialized software. By comparing successive images of the random speckle pattern acquired during the loading history on the surface of each specimen, DIC provided the full-field displacement and strain distributions.

## 3. Failure Criterion and Proposed Seawater Exposure Damage Model

This section is divided into three parts. Firstly, the Puck’s failure criterion is introduced. Then, the continuum dagame model used to predict the interlaminar progressive failure is addressed. The section ends with a proposed approach to simulate the effect of seawater on laminate.

### 3.1. Puck’s Failure Criterion

The initial failure theory was based on Puck’s failure criterion [47]. Puck’s criterion can identify the different fracture modes occurring in FRPs and quantify the repercussions of the lamina’s damage on the laminate. Furthermore, it can also predict the fracture plane’s orientation. Puck’s failure theory differentiates two types of fracture: Inter Fiber Fracture (IFF) and Fiber Fracture (FF). The IFF criteria are identified using stresses of the action plane since it is not possible to formulate a physically-based fracture criterion using the stresses based on the natural axes of the unidirectional (UD) composite [48]. Therefore, UD lamina stresses in the lamina coordinate system (1,2,3) are translated into the fracture plane coordinate system (1,n,t) using tensor transformation in the 2–3 plane:(1)$$\sigma_{n}\left(\theta \right)=\sigma_{2\;}cos^{2}\;\theta +\sigma_{3}\;sin^{2}\;\theta +2\tau_{23}\sin\theta \cos\theta$$
(2)$$\tau_{nt}\left(\theta \right)=-\sigma_{2}\sin\theta \cos\theta +\sigma_{3}\sin\theta \cos\theta_{\;}+\tau_{23}\left(cos^{2}\;\theta -sin^{2}\;\theta \right)$$
(3)$$\tau_{n1}\left(\theta \right)=\tau_{31\;}\sin\theta +\tau_{21}\cos\theta$$
where θ is the inclination of the fracture plane, $\sigma_{2\;}$ and $\sigma_{3}$ are normal stresses perpendicular to the fiber, and *τ*_21_, *τ*_31_ and *τ*_23_ are shear stresses.

The verification of IFF requires the previous identification of the fracture plane. Hereby, it is mandatory first to detect the plane’s fracture angle $\theta_{fp}$. The fracture plane is defined as the action plane with the maximum local stress exposure, and the local stress exposure, $f_{E}$, can be used as a direct measure of the fracture risk since it grows linearly with the stresses [49]. Therefore, $\theta_{fp}$ is searched numerically, calculating $f_{E}$ for all sections between $\theta =\left[-90^{{}^{\circ}};90^{{}^{\circ}}\right]$ to find the fracture plane. The local stress exposure can be obtained as follows:

For $\sigma_{n}\geq 0$,
(4)$$f_{E,IFF\;}\left(\theta \right)=\sqrt{{\left[\left(\frac{1}{R_{⊥}^{At}}-\frac{p_{⊥\psi }^{t}}{R_{⊥\psi }^{A}}\right)\sigma_{n}\left(\theta \right)\right]}^{2}+{\left(\frac{\tau_{nt}\left(\theta \right)}{R_{⊥⊥}^{A}}\right)}^{2}+{\left(\frac{\tau_{n1}\left(\theta \right)}{R_{⊥\|}^{A}}\right)}^{2}}+\frac{p_{⊥\psi }^{t}}{R_{⊥\psi }^{A}}\sigma_{n}\left(\theta \right)$$

For $\sigma_{n}<0$,
(5)$$f_{E,IFF\;}\left(\theta \right)=\sqrt{{\left(\frac{p_{⊥\psi }^{c}}{R_{⊥\psi }^{At}}\sigma_{n}\left(\theta \right)\right)}^{2}+{\left(\frac{\tau_{nt}\left(\theta \right)}{R_{⊥⊥}^{A}}\right)}^{2}+{\left(\frac{\tau_{n1}\left(\theta \right)}{R_{⊥⊥}^{A}}\right)}^{2}}+\frac{p_{⊥\psi }^{c}}{R_{⊥\psi }^{A}}\sigma_{n}\left(\theta \right)$$
where:(6)$$\frac{p_{⊥\psi }^{t}}{R_{⊥\psi }^{A}}=\frac{p_{⊥⊥}^{t}}{R_{⊥⊥}^{A}}\;cos^{2}\psi +\frac{p_{⊥\|}^{t}}{R_{⊥⊥}^{A}}\;sin^{2}\psi$$
(7)$$\frac{p_{⊥\psi }^{c}}{R_{⊥\psi }^{A}}=\frac{p_{⊥⊥}^{c}}{R_{⊥⊥}^{A}}\;cos^{2}\psi +\frac{p_{⊥\|}^{c}}{R_{⊥⊥}^{A}}\;sin^{2}\psi$$
(8)$$cos^{2}\psi =\frac{\tau_{nt}^{2}}{\tau_{nt}^{2}+\tau_{n1}^{2}}$$
(9)$$sin^{2}\psi =\frac{\tau_{nl}^{2}}{\tau_{nt}^{2}+\tau_{n1}^{2}}$$
(10)$$R_{⊥}^{A\;t}=R_{⊥}^{t}$$
(11)$$R_{⊥\|}^{A}=R_{⊥\|}$$
(12)$$R_{⊥⊥}^{A}=\frac{R_{⊥\|}}{2p_{⊥\|}^{c}}\left(\sqrt{1+2p_{⊥\|}^{c}\frac{R_{⊥}^{c}}{R_{⊥\|}}}-1\right)$$
where $R_{⊥⊥}^{A}$, $R_{⊥\|}^{A}$ or $R_{⊥}^{At}$ are the fracture resistances of the action plane to a single transverse shear stressing $\tau_{⊥⊥}$, longitudinal shear stressing $\tau_{⊥\|}\;$, or transverse tensile stressing $\sigma_{⊥}^{t}$, acting on the action plane, respectively. $R_{⊥\|}$ and $R_{⊥}^{t}$ are the longitudinal shear strength, and the transverse shear strength of the lamina, respectively, while $p_{⊥\|}^{t}$, $p_{⊥\|}^{c}$ $p_{⊥⊥}^{t},$ and $p_{⊥⊥}^{c}$ are experimentally determined inclination parameters provided by Puck et al. [50] and summarised in Table 1. After finding the fracture plane, Puck’s failure criterion can be verified with the following expressions:

Fiber failure in compression:(13)$$\frac{1}{-R_{\|}^{c}}\left[\sigma_{1}-\left(\nu_{⊥\|}-\nu_{⊥\|f}m_{\sigma_{f}}\frac{E_{\|}}{E_{\|f}}\right)\left(\sigma_{2}+\sigma_{3}\right)\right]=1$$

Fiber failure in tension:(14)$$\frac{1}{R_{\|}^{t}}\left[\sigma_{1}-\left(\nu_{⊥\|}-\nu_{⊥\|f}m_{\sigma_{f}}\frac{E_{\|}}{E_{\|f}}\right)\left(\sigma_{2}+\sigma_{3}\right)\right]=1$$

Matrix failure in compression ($\sigma_{n}<0$):(15)$$\;\;\;\;\;f_{E,IFF\;}\left(\theta \right)=1$$

Matrix failure in tension ($\sigma_{n}\geq 0)$:(16)$$\;\;\;\;\;f_{E,IFF\;}\left(\theta \right)=1$$
where $\nu_{⊥\|}\;$ is the major Poisson’s ratio of the fiber, $E_{\|}$ is the lamina longitudinal modulus parallel to the fibers, while $E_{\|f}\;$ is the longitudinal fiber modulus and $m_{\sigma_{f}}$ is the magnification factor for the transverse stress in the direction of the fiber. In this work, $m_{\sigma_{f}}$ was set to 1.3, as recommended for GFRP by Lee et al. [49]. Note that 1-direction corresponds to the fiber orientation, 2-direction is transverse in-plane orientation relative to the fiber direction, and 3-direction stands for transverse out-of-plane orientation.

### 3.2. Constitutive Relation and Progressive Failure Theory

Numerous progressive failure theories have been proposed to predict defect initiation and propagation in common materials. However, most damage models were designed for isotropic ductile materials, and their implementation is complex for anisotropic brittle materials such as FRP composites. Gu et al. [51] proposed a continuum damage model to predict the interlaminar progressive failure of CFRP laminates. A bilinear relation was used to describe the damage accumulation process. A good agreement between the experimental and numerical results was verified. In the present study, which is inspired in the work of Lee et al. [52], a stress-based Puck failure criterion was adopted to predict the damage progression. This approach is simple and widely used for strength estimation and progressive failure analysis.

The constitutive relation, which relates stress states to strain states, is written as follows [53]:(17)$$\left\{\begin{array}{c} \sigma_{1}\\ \sigma_{2}\\ \sigma_{3}\\ \tau_{12}\\ \tau_{23}\\ \tau_{31} \end{array}\right\}=\left[\begin{array}{cccccc} C_{11}^{} & C_{12}^{} & C_{13}^{} & 0 & 0 & 0\\ & C_{22}^{} & C_{23}^{} & 0 & 0 & 0\\ & & C_{33}^{} & 0 & 0 & 0\\ & & & 2G_{12} & 0 & 0\\ & Sym. & & & 2G_{23} & 0\\ & & & & & 2G_{31} \end{array}\right]\left\{\begin{array}{c} \varepsilon_{1}\\ \varepsilon_{2}\\ \varepsilon_{3}\\ \gamma_{12}\\ \gamma_{23}\\ \gamma_{31} \end{array}\right\}$$
where $\varepsilon_{I}$ and $\gamma_{\mathrm{ij}}$ represent three normal strains and three shear strains, respectively; I  and $\tau_{\mathrm{ij}}$ denote three normal stresses and three shear stresses, respectively, and $G_{\mathrm{ij}}$ denote the shear moduli. In addition, $C_{\mathrm{ij}}^{}$ can be defined with the Young’s moduli and Poisson’s ratios as will be established later in this section.

The constitutive relation parameters change based on the Element Weakening Method (EWM), if failure is verified [52]. The constitutive relation is modified by the damage parameters $d_{f}$ and $d_{m}\;$(Equations (18) and (19)) associated with the fiber and matrix failure, respectively. The variables $d_{ft}$, $d_{fc}$, $d_{mt}$, and $d_{mc}$ are related to fiber and matrix damage under tensile and comprehensive stress states, respectively. These variables assume the value of 1 if their associated failure mode occurs, as summarized in Table 2. All parameters for the constitutive relation, fiber and matrix damage variables, and material constants for the undamaged and damaged material stiffness matrix are written as follows:

(18)$$d_{f}=1-\left(1-d_{ft}\right)\left(1-d_{fc}\right)$$(19)$$d_{m}=1-\left(1-d_{mt}\right)\left(1-d_{mc}\right)$$(20)$$C_{11}^{}=\left(1-d_{f}\right)C_{11}^{0}$$(21)$$C_{22}^{}=\left(1-d_{f}\right)\left(1-d_{m}\right)C_{22}^{0}$$(22)$$C_{33}^{}=\left(1-d_{f}\right)\left(1-d_{m}\right)C_{33}^{0}$$(23)$$C_{12}^{}=\left(1-d_{f}\right)\left(1-d_{m}\right)C_{12}^{0}$$(24)$$C_{23}^{}=\left(1-d_{f}\right)\left(1-d_{m}\right)C_{23}^{0}$$(25)$$C_{13}^{}=\left(1-d_{f}\right)\left(1-d_{m}\right)C_{13}^{0}$$(26)$$G_{12}^{}=\left(1-d_{f}\right)\left(1-s_{mt}d_{mt}\right)\left(1-s_{mt}d_{mc}\right)G_{12}^{0}$$(27)$$G_{23}^{}=\left(1-d_{f}\right)\left(1-s_{mt}d_{mt}\right)\left(1-s_{mt}d_{mc}\right)G_{23}^{0}$$(28)$$G_{31}^{}=\left(1-d_{f}\right)\left(1-s_{mt}d_{mt}\right)\left(1-s_{mt}d_{mc}\right)G_{31}^{0}$$(29)$$C_{11}^{0}=E_{1}\left(1-\nu_{23}\nu_{32}\right)\delta$$(30)$$C_{22}^{0}=E_{2}\left(1-\nu_{13}\nu_{31}\right)\delta$$(31)$$C_{33}^{}=\left(1-d_{f}\right)\left(1-d_{m}\right)C_{33}^{0}$$(32)$$C_{12}^{0}=E_{1}\left(\nu_{21}+\nu_{31}\nu_{23}\right)\delta$$(33)$$C_{23}^{0}=E_{2}\left(\nu_{32}+\nu_{12}\nu_{31}\right)\delta$$(34)$$C_{13}^{0}=E_{1}\left(\nu_{31}+\nu_{21}\nu_{32}\right)\delta$$(35)$$\delta =1/\left(1-\nu_{12}\nu_{21}-\nu_{23}\nu_{32}-\nu_{31}\nu_{13}-2\nu_{21}\nu_{32}\nu_{31}\right)$$
where $C_{\mathrm{ij}}^{0}$ and $C_{\mathrm{ij}}^{}$ are the undamaged and damaged material stiffness components, respectively; $E_{1}$, $E_{2}$ and $E_{3}$ are Young’s moduli in the three laminae directions ($x_{1,}x_{2,}x_{3}$), respectively; $\nu_{\mathrm{ij}}$ are the Poisson’s ratios that can be defined as $\nu_{\mathrm{ij}}=-\varepsilon_{j}/\varepsilon_{i}$, with i≠j, and $G_{12}$, $G_{23}$ and $G_{31}$ are shear moduli in $x_{1}-x_{2}$, $x_{2}-x_{3}$ and $x_{3}-x_{1}$ planes, respectively. Lastly, $s_{mt}$ and $d_{mt}$ are the loss control factors for the shear stiffness caused by the matrix tensile and compressive failures, respectively. In this study, the used loss control factors values were: $s_{mt}=0.9$ and $s_{mt}=0.5$ as used in work by Seung Lee et al. [52].

### 3.3. Proposed Model for Seawater Damage

In the present study, a numerical 3D model for predicting the exposure damage to seawater was developed. The proposed approach states that the damage can be modelled by changing the lamina’s resistances as a function of the water concentration. Here, it is assumed that the water concentration of the finite elements closer to the permeable surface, $C_{CS}$, decreases linearly with the distance to the closer permeable surface, as the following equation suggests:(36)$$C_{CS}=1-\frac{D_{CS}}{D_{dist}}$$
where $D_{CS}$ is the Gaussian point’s distance to the closer permeable surface, and $D_{dist}$ is the diffusion distance. The lamina’s resistances parallel and perpendicular to fiber $R_{⊥,\|}^{t,c}$ are degraded as a function of the concentration:(37)$$R_{⊥,\|}^{t,c}=R_{⊥,\|}^{t,c}\left(1-kC_{CS}\right)$$
where *k* is an experimental fitting parameter. Concerning the increase in stiffness, Young’s modulus increases as a function of the normalized Fick’s law. Unlike the resistances, which can be a function of the concentration, Young’s moduli cannot, since it would create a fictitious stress concentration. Therefore, Young’s moduli are a function of the normalized Fick’s law as demonstrated in the following equation:(38)$$E_{i}=E_{0,i}\left(A+N_{Fl}B\right)\;$$
where $E_{i}$ is the Young’s moduli in the direction i, $E_{0,i}$ is the original Young’s moduli in the direction i; $N_{Fl}\;$ is the normalized Fick’s law, a function of the exposure time in days; and *A*, and *B* are fitting parameters.

The variation of the diffusion distance, $D_{dist}$, is based on Fick’s law. Hence, a linear relation between the weight gain, in Fick’s law, and the wet volume, $W_{volume}\left(D_{dist}\right)$, was assumed. The wet volume is the specimen volume affected by the diffusion and is a function of the diffusion distance. With this assumption, it is possible to find a function for the diffusion distance that forces the normalized wet volume to have the same behaviour as the normalized Fick’s law. Thus, it can be found a function for the diffusion distance that varies with the immersion days and is based on Fick’s law. When saturation is reached, the diffusion distance must be such that the wet volume equals the structure volume. Accordingly, the diffusion distance function is composed of two terms, as shown:(39)$$D_{dist}\left(t\right)=A\left(t\right)\times C\;$$
where A(t) is a continuum function that varies between [0,1] with the immersion time in days, t, which must be modified for the normalized wet volume to have the same behaviour as the normalized Fick’s law. C is the structural dimension, so that $W_{\mathrm{volume}}\left(D_{\mathrm{dist}}=C\right)=\mathrm{structure}\;\mathrm{volume}$. It should be noted that finding a function for the wet volume varying with the diffusion distance, $W_{volume}\left(D_{dist}\right)$ can be relatively tricky particularly for complex structures. However, if the wet volume increases linearly with the diffusion distance, then the diffusion distance can be a direct function of the normalized Fick’s law, i.e., A(t)=Normalized Fick’s law.

Many studies have shown that the weight gain in composite materials can be estimated using Fickian models [37,41,54,55]. Therefore, in this study, a one-dimensional Fickian model was used. The moisture absorption contends that $M_{t}$ is time-dependent and can be estimated by the following equation [56,57]:(40)$$M_{t}\left(t\right)=M_{\infty }\left\{1-\exp\left[-7.3{\left(\frac{D_{Coef}t\times 3600\times 24}{h{\;}^{2}}\right)}^{0.75}\right]\right\}\;$$
where $M_{\infty }$ is the maximum amount of absorption, *t* is time in days, h is the thickness, and $D_{Coef}$ is the diffusion coefficient. The following equation allows obtaining the diffusion coefficient $D_{Coef}$, using two points at time $t_{1}$ and $t_{2}$ from the linear part of Fickian diffusion curve, i.e.,
(41)$$D=\pi {\left(\frac{hk}{4M_{\infty }}\right)}^{2}\;$$
where k can be obtained as suggested by Wang et al. [14]:(42)$$k={\left(\frac{M_{1}-M_{2}}{\sqrt{t_{1}}-\sqrt{t_{2}}}\right)}^{}\;$$
where $M_{1}$ and $M_{2}$ are the percentage of water absorbed at time $t_{1}$ and $t_{2}$, respectively.

The parameters are summarised in Table 3, and the comparison between the fitted Fick’s law and the experimental results is displayed in Figure 1. Overall, as can be seen in the figure, the proposed function is close to the experiments regardless of the immersion time. The maximum differences, which were found for immersion times of 640 days, were lower than 8.8%.

## 4. Finite-Element Based Implementation

The iterative procedure implemented in this study to simulate the effect of seawater exposure on mechanical properties in FGRPs was developed using the Abaqus commercial software package along with a UMAT subroutine written in Fortran. The algorithm framework is presented in Figure 2. As can be seen in the figure, at each time increment for each integration point, the information on material properties, strain tensor, strain incrementation and state variables are passed to the UMAT subroutine. After sending that information, if the iteration is the first one, the waterdamage routine is called. This condition reduces simulation time since this routine requires data that do not vary with the increment number. Concisely, the routine verifies if the integration point is within the zone damaged by the seawater, and if so, the tensile and compressive strengths parallel and perpendicular to fiber, $R_{\|,⊥}^{t,c},$ are degraded.

Then, the constitutive matrix is evaluated, and following that, the stress tensor is calculated. In the next step, the SearchFP routine is called, which calculates the value of $\theta_{fp}$ (see Section 3.1). After that, both the IFF and the FF failure criteria are verified (see Section 3.1). The respective degradation law is applied if a failure occurs. Since the constitutive matrix can change in the case of failure, the failure criteria must be verified again. Finally, the state variables are updated, and the updated constitutive matrix, the stress tensor, and the state variables are sent to Abaqus. Convergence is achieved if the residual force is lower than the tolerance value. If it is higher, the increment is repeated with a lower increment time to ensure easier convergence.

The finite element mesh of the tested geometry, see Figure 3, was generated with one element through-thickness for each lamina using 8-node linear hexahedral elements with reduced integration (C3D8R). The assembled model had 123,262 elements and 146,874 nodes. This element type uses a single integration point located at the centroid of the element which provides excellent benefits, since the number of evaluations during the simulation is substantially reduced. However, the difference between the number of nodes and the number of integration points can lead to mesh instability, known as hourglassing. Hence, the enhanced hourglass control option was activated for all simulations.

Mesh convergence is another critical subject in any finite element analysis. In general, mesh convergence problems arise in regions with severe geometric discontinuities [58]. To prevent this concern, the element size around the hole was reduced to a 0.1 mm side, ensuring convergence in terms of stress and strain. In remote positions, the element size was larger to reduce the computational overhead. Regarding the type of finite element analysis, although a quasi-static analysis is particularly stable for large time steps, using lower time steps can lead to improved solutions. However, it can result in higher simulation times. Thus, each simulation was executed with a time step of 0.01 s, while the total time of 1 s was simulated. The material properties used in the simulations were those recommended by the WWFE [59].

The boundary conditions used to simulate the tensile tests were adopted using kinematic constraints in zones A and B, represented in Figure 3. In the former zone, all translations and rotations were restrained, while in the latter zone only the translation on the *x*-axis direction was allowed. The loading was applied on constraint B in the *x*-axis direction.

## 5. Results and Discussion

The first objective of the present study was to validate the implementation of Puck’s failure theory. Thus, the strain fields near the hole region simulated with the proposed approach were compared with those obtained in the experimental tests for different loading levels [60]. The experimental results, as referred to above, were measured using Digital Image Correlation (DIC) [61,62]. The comparative analysis was carried out using the first principal strain. As depicted in Figure 4, the numerical simulations and experimental results agree well, regarding the strain fields. In fact, when comparing DIC results to the implemented Puck’s failure model, the strain profile is similar for the three loading levels which validates the proposed approach.

Furthermore, although during the tensile tests loads were uniformly applied on the external surfaces of the specimen during the tensile tests, from the experimental results, it is possible to observe that the first principal strain field at the hole surface is not symmetrical. Puck’s failure theory can predict this asymmetrical behaviour, which is likely to happen due to the laminae sequence, resulting in an asymmetrical matrix damage evolution. Figure 5 compares the maximum first principal strain with the matrix damage as load increases. At 3.0 kN or lower, no matrix damage occurs at the surface and no asymmetry is perceptible in the strain field. As the load increases, matrix damage starts to build up, and the strain asymmetry is lightly perceptible. At 5.0 kN or higher, a pronounced asymmetrical matrix damage pattern occurs, generating the uneven strain pattern that is verified in the experiments.

After the validation of the strain fields near the hole region, which is based on the implemented Puck’s failure theory, the tensile stress–strain tests for the control samples, i.e., the specimens not immersed in seawater, were also compared with the numerical simulations. The maximum load and displacement at failure for the tensile tests for the control and the immersed specimens are summarized in Table 4 and Table 5, respectively. The results for the maximum load are, for all the tests, relatively close to the expected value; however, the maximum displacement results revealed some scatter. The stress–strain plots determined numerically and obtained in the experiments are shown in Figure 6. In all simulations, the final failure is assumed to occur when the fiber failure crosses all the specimen width, accompanied by a sudden escalation of the displacement.

The numerical model predicts the failure stress with an error of 8.3% and the strain at failure with an error of 10.7% compared with the average experimental results. Even though the model numerical model predicts the final values with reasonable precision, the stress–strain plot exhibits a nonlinear behaviour at a stress of approximately 80 MPa and then resumes the initial linear behaviour until the total failure. Although there is a slight nonlinearity on the experimental curve, it is noticed a small disagreement between the experimental data and the numerical behaviours. This nonlinear behaviour is caused by the first IFF (see Section 3.1), which is documented in the work of Knops [49].

The numerical model can also provide important insights into the fundamental mechanisms associated with the damage development. The experimental evidence, as already verified by Gonçalves [60] and Aguiar [63], suggests that the borderline of the hole is where the cracks start to spread, which is substantiated by the numerical simulations. Figure 7 and Figure 8 depict the matrix and fiber damage progression in the hole region for applied loads of 7.0 kN, 8.0 kN, 9.0 kN, and 9.7 kN, at the surface and through-the-thickness. The matrix failure, see Figure 7, precedes the fiber failure, as expected. It can be observed at 8.0 kN that, as the load increases and the fiber damage develops, the matrix damage exhibits a rapid growth. At 9.0 kN, the matrix damage covers all the width and thickness in the middle section of the specimen.

Regarding the fiber failure, as can be distinguished in Figure 8, it only takes place on the 0° plies, since the other pliers’ fibers are not being loaded. At about 7.0 kN, the fiber damage appears, and then, as the load increases, it propagates along a 90° angle relative to the loading direction. At last, the test specimen is fully fractured under approximately 9.7 kN, and it is clear that the failed region induced by IFF is much wider than that induced by FF.

Figure 8 also shows the uniaxial stress–strain curves obtained experimentally and simulated numerically for 900 days of immersion considering the methodology proposed to account for the damage caused by exposure to seawater. Overall, both curves are relatively close in the entire range. The numerical model predicts the failure stress with an error of 2% and the displacement at failure with an error of 4.9%. Furthermore, the initial nonlinear behaviour of the curve, already distinguished for the specimens not immersed in seawater, is also visible in this case. Similarly, it is also softened with the decrease in the ductility of the tested composite. As can be seen in Figure 9, the mechanical behaviour of the laminate is accurately predicted for applied stresses higher than 80 MPa, which is an interesting outcome and demonstrates the adequacy of the proposed methodology.

Figure 9 illustrates the seawater concentration after 30, 60, 150, 400 and 900 immersion days. It can be noticed that, with the increase of the immersion days, the diffusion distance increases, causing the diffused regions to cover the specimen’s volume to a greater extent. It is interesting to observe that the concentration is identical from day 400 to 900, although with more than twice the exposure time. This result is expected since the weight gain in the last section of Fick’s law remains approximately constant.

Figure 10 displays the stress–strain plots simulated with the proposed numerical model for immersion days in the range 0 to 900 days. It is clear that most of the strength and ductility loss occurs in the first days of immersion. The differences between the cases corresponding to 300 and 900 days of immersion are not significant, particularly with regard to the maximum stress; regarding the strain at failure, although the results are not much different, there is a more relevant reduction with the increase of the immersion time. Nevertheless, since most of the water absorption occurs in the first months of immersion, see Figure 1, it is expected that most of the damage also develops during this period.

The relationship between the maximum applied stress and the immersion time obtained by means of the numerical model developed in this study is exhibited in Figure 11. As inferred above, it can be noticed that most of the damage occurs in the first three months of immersion, and after approximately 250 days, no significant changes occur in the failure stress. Based on the numerical simulations, the dialectical relationship between the failure stress and the immersion time was established via nonlinear regression:(43)$$\;\;\;\;\;S\left[\mathrm{MPa}\right]=244.58{\left(1+ID\right)}^{-4.0292}$$
where *S* is the failure stress, and *ID* is the number of immersion days. Figure 11 also plots the experimental values of the failure stress obtained for 0 days and 900 days of immersion. In this representation, it can also be observed that the simulated results agree well with the experiments.

Another important application of the proposed methodology is the design improvement of engineering components subjected to hostile environmental media. For example, we can consider different surfaces as permeable, or impermeable, and simulate the effect of these changes on structural response. This can be extremely important to increase the long-term durability because surface protective coatings degenerate over time and need to be renewed. Since diffusion phenomena can happen on different surfaces and can have different effects on mechanical strength, the numerical model developed in this study is a valuable tool to identify a priori the most critical surfaces allowing an optimized design to be achieved in a simple and fast manner, but also allows to simulate the structural behaviour and quantify eventual benefits associated with different combinations of impermeable and permeable surfaces.

Figure 12 compares the seawater concentration considering all permeable surfaces (Figure 12a), impermeable lateral surfaces (Figure 12b), and an impermeable hole surface (Figure 12c). As can be seen in the figure, the protection against damage by seawater of the hole surface significantly reduces the seawater concentration in the central region around the hole (Figure 12c). On the contrary, in the case of the hole surface not being protected and the lateral surfaces being simultaneously protected (Figure 12b), the seawater concentration is significantly higher around the hole region even though it is necessary to protect a much wider area. When no protection is used, high seawater concentrations are found not only at the lateral surfaces but also at the hole region.

The different seawater concentration distributions reported above are likely to affect the mechanical behaviour of the laminate. Figure 13 compares the uniaxial stress–strain response considering the non-permeability of the lateral surfaces and the non-permeability of the hole surface for 50, 150 and 900 immersion days. Comparing the model predictions between the simultaneously impermeable lateral surfaces (165 mm × 2.3 mm) and the impermeable hole surface, the difference in failure stress is very significant for the three immersion periods. For 50, 150 and 900 immersion days, the numerical simulations indicate that the specimen can sustain 10.4, 16.2 and 11.8% more load before failure if the hole surface is protected, respectively. This is a coherent outcome since, with an impermeable hole surface, no reduction in mechanical properties occurs near the hole where the fiber damage initiates (see Figure 8). As a result, the fiber damage only initiates at higher stresses enabling the specimen to support more load. In terms of structural integrity, when statically loading the specimen, protecting the hole surface is more efficient than protecting the lateral surfaces, even though the area of the lateral surfaces is more than 21 times the area of the hole surface.

Thus, an efficient way to improve the structural strength in a marine environment for structural components made of GFRPs and having holes would be the application of protective coatings on the hole surface.

## 6. Conclusions

This paper presented a 3D numerical procedure for predicting the effect of seawater exposure on mechanical behaviour of glass fiber-reinforced polymers. The following conclusions can be drawn:The linear relation between the weight gain and the wet volume was adequate to model the diffusion process. The maximum errors between the fitted function and the experimental results were lower than 9%.The first principal strain fields near the hole surface simulated with the proposed methodology were quite similar to those measured in the experiments for the different loading cases with digital image correlation;The model predicted both the maximum stress and the strain at failure with good accuracy, irrespective of the immersion time. In the absence of seawater, the results were more accurate;The stress–strain plots exhibited a nonlinear behaviour. However, this non-realistic behaviour only occurred in the early stage of the loading process. After a certain stress level, the numerical simulations and the experimental results behaved similarly;A power relationship between the stress at failure and the immersion time was found. Most of the damage caused by the exposure to seawater occurred in the first three months;The model can help designers to identify the areas where the application of impermeable coatings can improve the resistance to seawater exposure. The cases analysed showed gains in maximum stress higher than 10%.

## Figures and Tables

**Figure 1 polymers-14-03955-f001:**
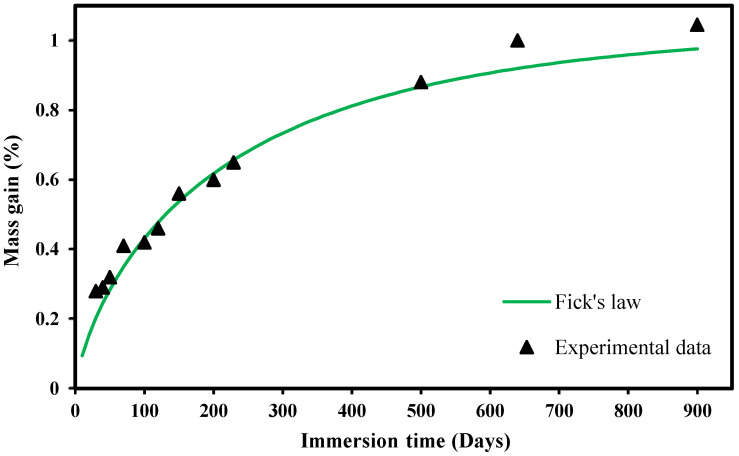
Comparison of Fick’s law for predicting the seawater absorption and the experimental results for the tested laminate.

**Figure 2 polymers-14-03955-f002:**
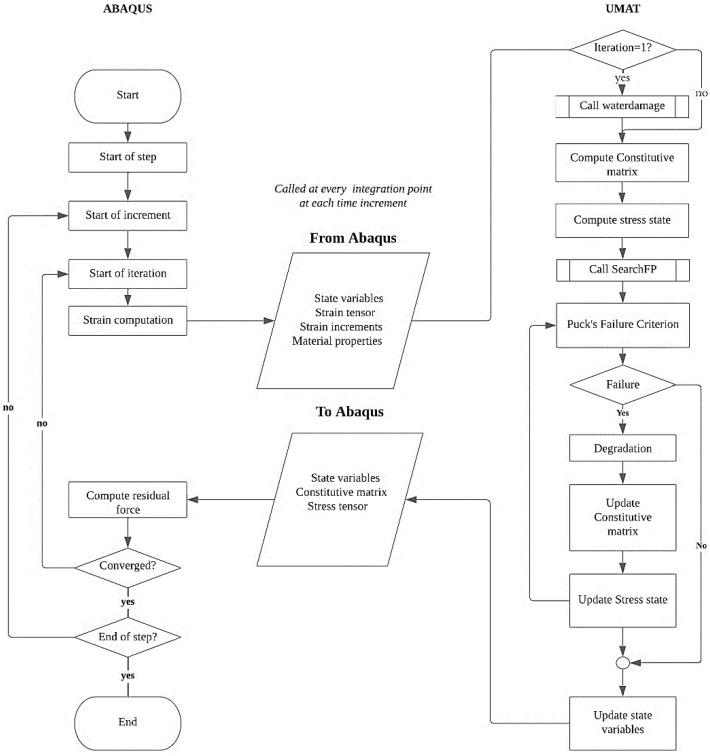
Algorithm of the 3D-FE iterative procedure developed in this study.

**Figure 3 polymers-14-03955-f003:**
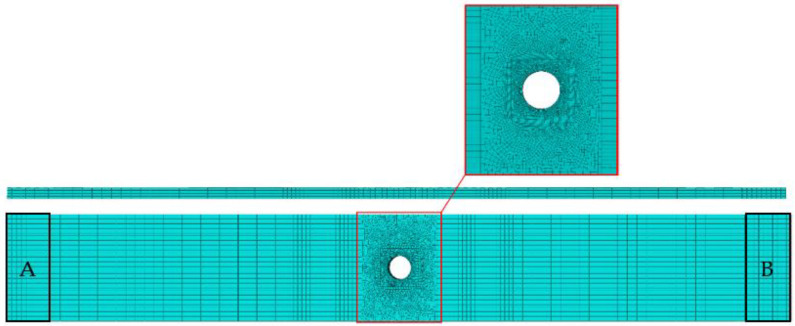
Finite element mesh used in the numerical simulations.

**Figure 4 polymers-14-03955-f004:**
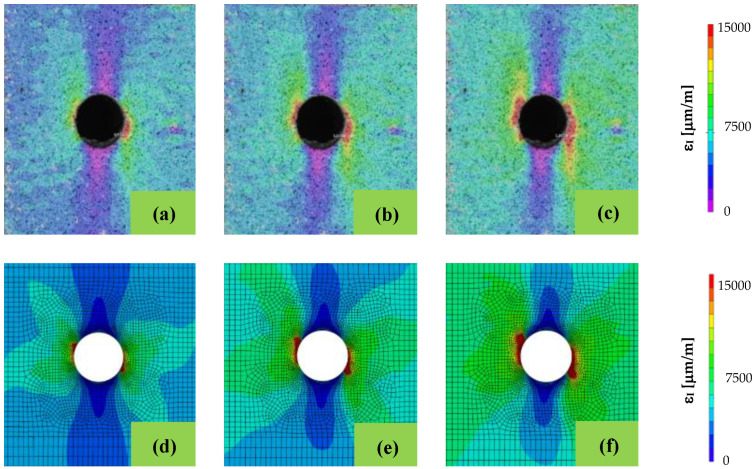
Comparison between the first principal strain (ε_I_) fields obtained experimentally and numerically. Images a, b and c correspond to the experimental results, and the remaining ones to the simulations with the application of Puck’s failure theory; the images (**a**,**d**), (**b**,**e**) and (**c**,**f**) correspond to loads of 4.3 kN, 4.875 kN and 5.5 kN, respectively.

**Figure 5 polymers-14-03955-f005:**
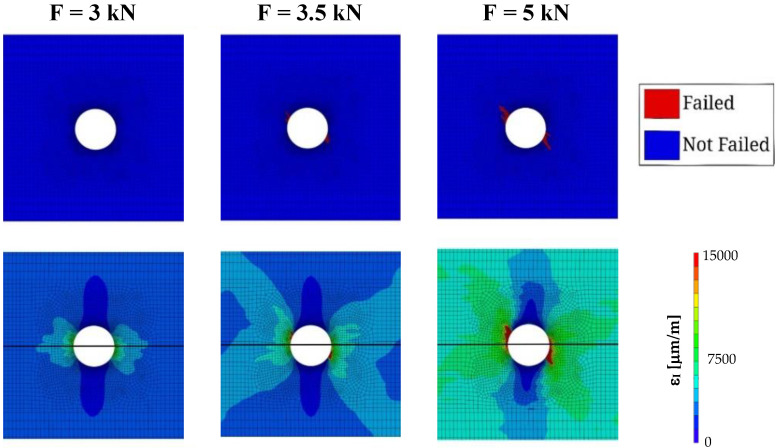
Matrix damage (IFF) and first principal strain (ε_I_) pattern progression.

**Figure 6 polymers-14-03955-f006:**
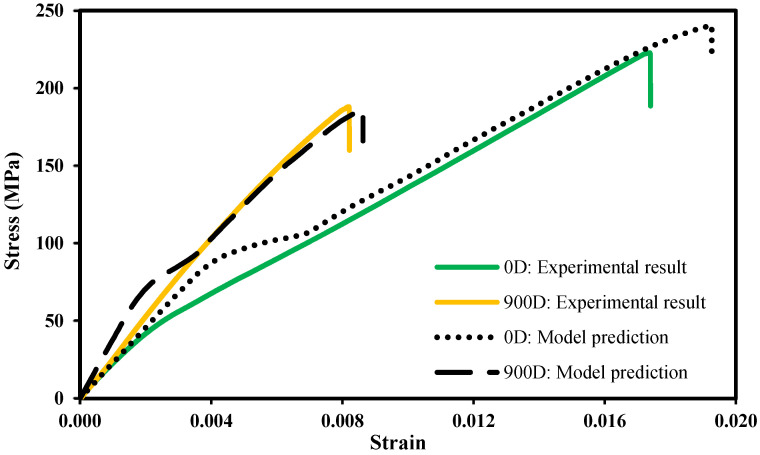
Stress–strain plot: experimental results and prediction of the effects of the seawater exposure for 0 and 900 immersion days.

**Figure 7 polymers-14-03955-f007:**
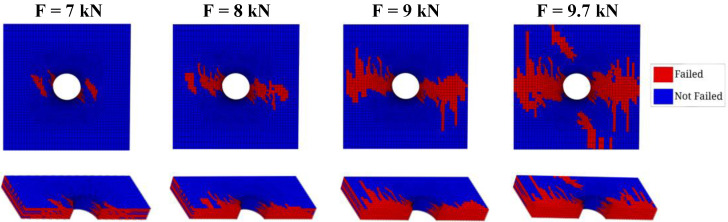
IFF pattern progression prediction.

**Figure 8 polymers-14-03955-f008:**
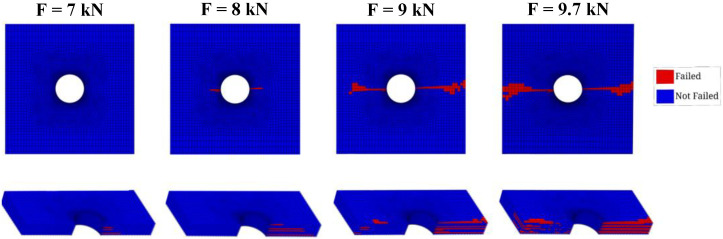
FF pattern progression prediction.

**Figure 9 polymers-14-03955-f009:**
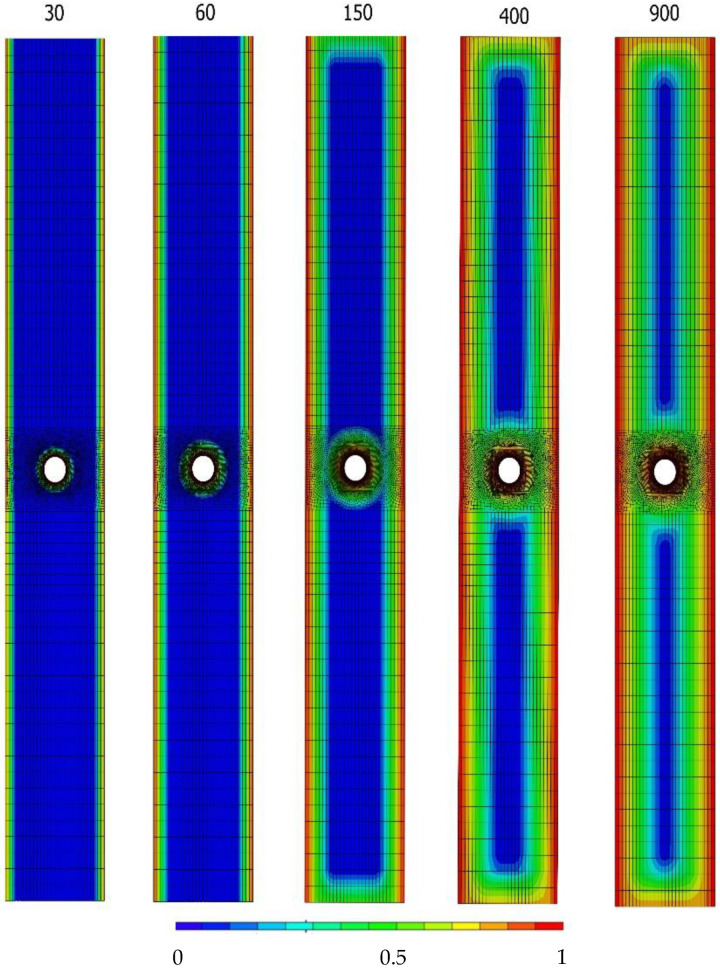
Prediction of the concentration of seawater after 30, 60, 150, 400 and 900 days of immersion.

**Figure 10 polymers-14-03955-f010:**
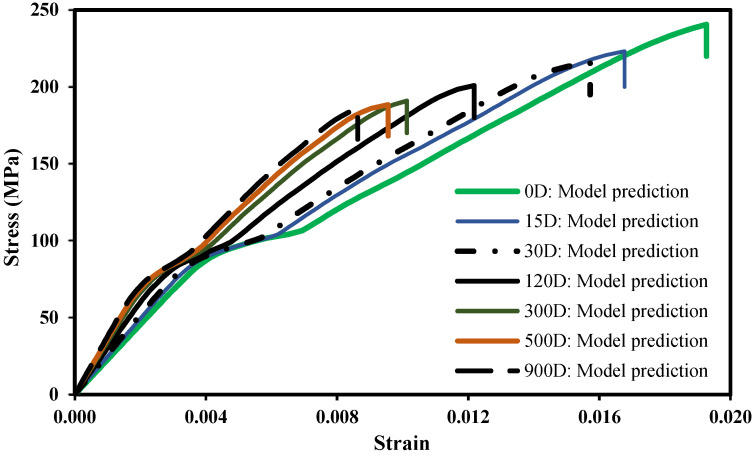
Stress–strain plot: model prediction of the effects of the seawater exposure for different immersion time.

**Figure 11 polymers-14-03955-f011:**
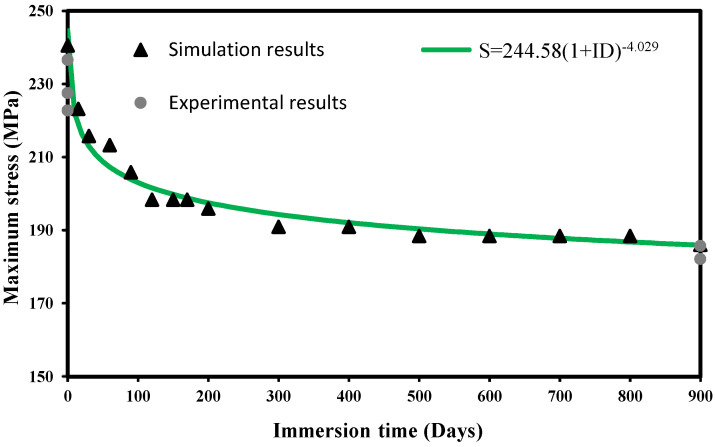
Maximum stress against immersion time: numerical predictions and experimental results for 0 and 900 immersion days.

**Figure 12 polymers-14-03955-f012:**
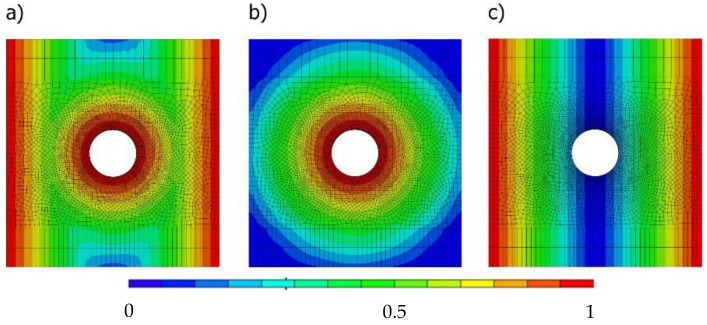
Seawater concentration after 900 immersion days with: (**a**) all permeable surfaces; (**b**) impermeable lateral surfaces; (**c**) impermeable hole surface.

**Figure 13 polymers-14-03955-f013:**
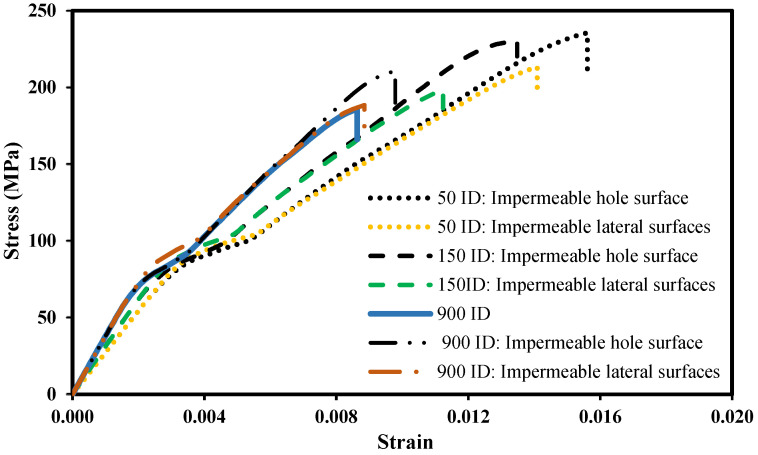
Prediction of the maximum stress for different impermeable surfaces.

**Table 1 polymers-14-03955-t001:** Recommended inclination parameters for GFRP, adapted from [50], Composites Science and Technology, 2002.

$p_{⊥\|}^{t}$	$p_{⊥\|}^{c}$	$p_{⊥⊥\;}^{t}\;and\;p_{⊥⊥}^{c}$
0.30	0.25	0.20 to 0.25

**Table 2 polymers-14-03955-t002:** Damage values for each failure mode.

Failure Mode	Damage Value
Tensile fiber	$d_{ft}=1$
Compressive fiber	$d_{fc}=1$
Tensile matrix	$d_{mt}=1$
Compressive matrix	$d_{mc}=1$

**Table 3 polymers-14-03955-t003:** Parameters of water absorption behaviour for the application of Fick’s law.

$\mathrm{Saturation}\;\mathrm{Absorption}\;M_{\infty }(\%)$	Slope *k* $\left(\times 10^{-5}d\right)$	Thickness *h* (mm)	Mass Diffusivity Coefficient *D* $\left(\times 10^{-8}\;mm^{2}/d\right)$
1.04	14.35	2.3	2.31

**Table 4 polymers-14-03955-t004:** Maximum load and displacement at failure for control specimens (not immersed), adapted from [60,62,63].

Nº	Maximum Load, F (N)	Maximum Displacement, δ (mm)
1	8969	2.87
2	9545	3.19
3	9155	2.33
Average	9223	2.8
St. Dev.	208	0.4

**Table 5 polymers-14-03955-t005:** Maximum load and displacement at failure for specimens immersed for 900 days, adapted from [60,62,63].

Nº	Maximum Load, F (N)	Maximum Displacement, δ (mm)
1	8380	2.69
2	7566	1.35
3	7475	2.16
Average	7807	2.1
St. Dev.	407	0.6

## Data Availability

The raw data presented in this study are available on request from the corresponding author.

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
