# Peer review of "Numerical Modeling of Damage Caused by Seawater Exposure on Mechanical Strength in Fiber-Reinforced Polymer Composites"

_polymers, 2022, doi:10.3390/polym14193955_

Round 1

Reviewer 1 Report

A very interesting study was conducted on the effect of seawater exposure on the mechanical properties of composites through experimental and simulation. Overall, the research work was well designed and some interesting results were obtained. To further improve the quality of the paper, the comments below should be replied. 

1# Abstract, please provide some quantitative analysis results and conclusions about finite element analysis. The consistency (R2) between simulation results and experimental results should be further indicated. In addition to the stress-strain curve, some other simulation results, such as the stress distribution and concentration, should also be presented in the current abstract.

2# Introduction, when mentioning the properties and advantages of FRP, it is suggested to analyze and summarize different kinds of FRPs in engineering application. This is because the performance and advantages of different FRPs (CFRP, GFRP and BFRP) have significant differences. For example, CFRP has excellent mechanical properties, fatigue properties and corrosion resistance with higher prices and lower elongation at break. In contrast, GFRP and BFRP have good mechanical properties, but their long-term service performance is more fragile especially in the alkaline environment of concrete. Please review the latest research work for necessary supplement. Materials and Structures, 2020, 53: 73. Composite Structures, 2019, 229: 111427.

3# As for the long-term durability, lots of research work has been done, including the water absorption, long-term evolution, degradation mechanism and prediction model of FRP exposed in the service environments. At present, the summary of durability in the introduction is not sufficient, please refer to some research work on durability by R. Guo and GJ. Xian to supplement relevant evolution mechanism (DIC technology) and life prediction of FRPs.

4# Please provide the intervals for water absorption tests and tensile tests. How many specimens were used for tensile testing?

5# Why dnot test the thermal and microscopic properties of FRPs after aging? These tests are very important to reveal the degradation mechanism of FRP.

6# In part 3 and 4, for the proposed model for seawater damage model, how to consider the influence of salt ions in seawater? If you don’t consider this condition, what is the difference between seawater environment and distilled water environment?

7# In figure 1, Fick’s law was adopted for predicting the seawater absorption. Please provide the relevant formula of Fick’s law before this.

8# Some references in the paper show errors, and it is suggested that the authors check the full text and modify it.

9# The stress-strain curve in the figure 10 is predicted by the model. What is the agreement between the predicted result of the model and the experimental stress-strain curve?

10# Does the fitting parameter of formula 41 have certain physical significance? Why choose such a model to fit the strength-time?

11# The conclusions should be further refined to include 3-4 key points.

Reviewer 2 Report

This work is devoted to the construction of numerical models of the effect of sea water on the mechanical strength of fiber-reinforced polymer composites. The article includes a large amount of calculated values, their visualization, etc. The introduction quite clearly introduces the problems of this study. The relevance of the study is beyond doubt. A good article with a fairly well-developed mathematical part is an important contribution to modern science. However, there are some points for minor improvement:

1. Read the article carefully before resubmitting. Perhaps there is a bug in the program that links to the literature. This is indicated by the phrase "Error! Reference source not found.", which appears frequently in the text.

2. In addition, more comparisons with analogues from the literature can be indicated in the text. This will give more validity to the conclusions of the authors.

3. It is desirable to cite the article: 10.1007/s13399-021-01895-y.

4. Conclusions can be made more concise.

5. It is desirable for the authors to indicate in more detail the practical significance of the results of this study.

Round 2

Reviewer 1 Report

Paper can be accepted in the current form.

Author Response

Thank you very much.
